# Tissue distribution of *Coxiella burnetii* and antibody responses in macropods co-grazing with livestock in Queensland, Australia

**Anita Tolpinrud**[1]*, **Mythili Tadepalli**[2], **John Stenos**[2], **Louis Lignereux**[3], **Anne-Lise Chaber**[3], **Joanne M. Devlin**[1], **Charles Caraguel**[3°], **Mark A. Stevenson**[1°]

**1** Asia Pacific Centre for Animal Health, Melbourne Veterinary School, The University of Melbourne, Parkville, Victoria, Australia, **2** Australian Rickettsial Reference Laboratory, University Hospital Geelong, Geelong, Victoria, Australia, **3** School of Animal and Veterinary Sciences, The University of Adelaide, Roseworthy, South Australia, Australia

° These authors contributed equally to this work.

* atolpinrud@gmail.com

**Data Availability Statement:** All data for this project are available in the Open Science

## Abstract

*Coxiella burnetii*, the causative agent of Q fever, is a zoonotic bacteria of global public health significance. The organism has a complex, diverse, and relatively poorly understood animal reservoir but there is increasing evidence that macropods play some part in the epidemiology of Q fever in Australia. The aim of this cross-sectional survey was to estimate the animal- and tissue-level prevalence of coxiellosis amongst eastern grey (*Macropus giganteus*) and red (*Osphranter rufus*) kangaroos co-grazing with domestic cattle in a Q fever endemic area in Queensland. Serum, faeces and tissue samples from a range of organs were collected from 50 kangaroos. A total of 537 tissue samples were tested by real-time PCR, of which 99 specimens from 42 kangaroos (84% of animals, 95% confidence interval [CI], 71% to 93%) were positive for the *C. burnetii* IS*1111* gene when tested in duplicate. Twenty of these specimens from 16 kangaroos (32%, 95% CI 20% to 47%) were also positive for the *com1* or *htpAB* genes. Serum antibodies were present in 24 (57%, 95% CI 41% to 72%) of the PCR positive animals. There was no statistically significant difference in PCR positivity between organs and no single sample type consistently identified *C. burnetii* positive kangaroos. The results from this study identify a high apparent prevalence of *C. burnetii* amongst macropods in the study area, albeit seemingly with an inconsistent distribution within tissues and in relatively small quantities, often verging on the limits of detection. We recommend Q fever surveillance in macropods should involve a combination of serosurveys and molecular testing to increase chances of detection in a population, noting that a range of tissues would likely need to be sampled to confirm the diagnosis in a suspect positive animal.

## Introduction

*Coxiella burnetii*, the aetiological agent of Q fever, is a zoonotic coccobacillus capable of infecting a wide range of vertebrate and arthropod hosts. The disease was first described in the late

Framework at https://osf.io/wd9u3 (DOI: 10.17605/
OSF.IO/WD9U3).

**Funding:** This research was supported by an
Australian Government Research Training Program
(RTP) Scholarship (AT) and was carried out as part
of the Taking the Q (query) out of Q fever project,
supported by funding from the Australian
Government Department of Agriculture, Fisheries
and Forestry (URL: https://www.agriculture.gov.au/
) as part of its Rural R&D for Profit program (grant
RnD4Profit-15-02-008, MS/JS/CC), administered
through AgriFutures Australia (Rural Industries
Research Development Corp). The funders were
not involved in the study design, data collection
and analysis, decision to publish, or preparation of
the manuscript.

**Competing interests:** The authors have declared
that no competing interests exist.

1930s in Queensland, Australia, and the pathogen has since been detected worldwide, except in New Zealand [1,2]. *C. burnetii* is an obligate intracellular pathogen and resides and reproduces within host macrophages and monocytes [3]. In pregnant placental mammals, the pathogen can also be found in particularly high density in placental tissue, where it colonises trophoblasts [4,5]. Acute infection in humans usually manifests as a self-limiting flu-like illness, although it can sometimes present as severe or life-threatening pneumonia or hepatitis. In a small percentage of cases, *C. burnetii* can persist for years and cause serious chronic illness, including endocarditis. Animal infections, referred to as coxiellosis, are usually inapparent, although in some cases they can present as severe placentitis and reproductive failure in placental mammals, leading to late-term abortions and stillbirths [2]. Despite being an intracellular pathogen, *C. burnetii* can enter an environmentally stable, nonreplicating spore-like form that is shed in milk, urine, faeces, semen, vaginal secretions, and birth products [6–9]. The bacterium is highly infectious and can persist in the environment for months or years. It is most commonly spread by inhalation of contaminated aerosols, although less frequent transmission routes include ticks, ingestion, or sexual transmission [8].

Domestic ruminants are generally considered the main animal reservoir of *C. burnetii* infection for humans [8,10]. Infected ruminants can shed large numbers of bacteria in their birth products, and proximity to humans, high stocking densities and the frequently intensive nature of livestock farming promote transmission [11–13]. However, there is compelling evidence that wildlife and ticks contribute to the maintenance and transmission of *C. burnetii* to varying extents, although the epidemiology as it relates to this sylvatic cycle remains relatively poorly understood [14].

Q fever is endemic in many parts of Australia, where it is a notifiable disease in humans. Although most notifications are caused by contact with infected domestic animals, a growing number of human cases of Q fever in Australia have been linked to non-typical animal exposures, such as feral animals and wildlife [15,16]. Kangaroos (family Macropodidae) and bandicoots (family Peramelidae) have often been implicated as the most likely primary wildlife reservoirs of Q fever in Australia and there is mounting circumstantial evidence that macropods are the likely source of some human infections [17–21]. These endemic marsupials occupy diverse habitats, ranging from arid deserts to temperate forests and are often found in close proximity to humans and livestock. Macropods, in particular, frequently share grazing land with domestic livestock, which provides ample opportunities for potential pathogen transmission. Studies have reported that kangaroos often have a high seroprevalence of antibodies to *C. burnetii*, averaging around 30% and sometimes exceeding 80% in certain populations [18,22–26]. Retrospective investigations of human Q fever case series in New South Wales (NSW) and Queensland, both areas with some of the highest notification rates in the country, found that up to 43% of cases reported direct or indirect contact with macropods before the onset of illness, although many of these also reported concurrent contact with livestock [16,21,27–29].

Despite a body of largely circumstantial evidence linking macropods to human infection, relatively little work has been done to elucidate the epidemiology of *C. burnetii* in macropods, which is likely to differ from placental mammals. Only a few studies have attempted to determine the molecular prevalence in these species, primarily in faeces and blood. Between 4 and 25% of faecal samples from western grey kangaroos (*Macropus fuliginosus*) have been found to be positive for *Coxiella* DNA, providing some evidence that faecal shedding is a possible route of transmission [23,24]. Aerosolisation of contaminated animal faeces, for instance through mowing or brush cutting, has been implicated as the likely source of previous human outbreaks of Q fever and this is also thought to be a risk with kangaroo faeces [9,30,31]. Additionally, the bacterium has also been detected by PCR in 6 out of 17 (35%) blood samples from

eastern grey kangaroos in northern Queensland, indicating that active bacteraemia could be a feature of infection, although isolation of viable bacteria from kangaroo blood has been reported only once [18,22]. More recently, PCR analyses of 47 raw kangaroo meat packages sold as pet food found that half contained *C. burnetii* DNA [32]. Genotyping of the positive samples identified three genotypes all belonging to previously observed isolates from human Q fever patients [32]. However, as pet meat consists of a mix of offal and muscle, possibly from more than one animal, determining the animal-level prevalence or identifying the source of the organism within the animal was not possible. A better understanding of the disease ecology and tissue tropism of coxiellosis in macropods would be beneficial to aid with surveillance, infection control and risk mitigation.

The objective of this cross-sectional survey was to estimate the animal- and tissue-level molecular prevalence of *C. burnetii* in macropods co-grazing with domestic cattle in a Q fever endemic area in Queensland. Additionally, we aimed to describe the relationship between seroprevalence and molecular prevalence within the study population. Although the overall seroprevalence for this population has been reported elsewhere [26], this paper provides a more detailed analysis of the serological results in the context of the molecular results and demographics. In this way, we hope to improve our understanding of the epidemiology of *C. burnetii* infections in macropods and provide recommendations in relation to which tissues should be tested to return the highest probability of *C. burnetii* detection.

## Materials and methods

### Ethics

Samples for this study were collected opportunistically from macropods that were shot and killed by licensed kangaroo shooters under the relevant landholder's damage mitigation permit. The collection of scavenged tissues from animals killed for purposes other than scientific research does not require prior ethics approval, however, the University of Queensland Office of Research Ethics was notified (project code SVS/489/18) and sample collection was performed under section 57 of the *Nature Conservation (Wildlife Management) Regulation 2006*, as laid out by the Queensland state government. The authors confirm that the ethical policies of the journal, as noted on the journal's author guidelines page, have been adhered to.

### Sampling site and specimen collection

A cross-sectional survey of *C. burnetii* prevalence among eastern grey (*Macropus giganteus*) and red (*Osphranter rufus*) kangaroos was conducted on a beef cattle station southwest of Roma, South West Queensland (27˚15'54" S, 148˚04'53" E) in May 2019. The climate in this area of Queensland is classed as hot semi-arid (BSh) by the Köppen climate classification [33] and the landscape is predominantly alluvial plains with remnant dry eucalypt woodlands fragmented by agriculture [34,35]. Cattle and sheep are the main livestock grazed in this area [36]. Coxiellosis is endemic in cattle in the region, and human cases of Q fever are frequently reported [28,37].

Kangaroos that were shot as part of an ongoing culling program separate from this study were sampled opportunistically (n = 50) over a three day period. The animals were killed by licenced kangaroo shooters and the carcasses were brought to a central processing site for specimen collection, which was carried out within six hours after death. Samples of the heart, lung, liver, spleen, mediastinal and mesenteric lymph nodes, and reproductive organs (uterus and pouch from females; testes and epididymis from males) were aseptically collected into individually labelled ziplock bags and frozen at -20˚C until processing. To minimise the risk of DNA cross-contamination between organs and kangaroos, a three-step wet disinfection

method was employed: stainless steel dissection instruments were immersed in a 1% sodium hypochlorite solution (household bleach), followed by 70% ethanol to neutralise the sodium hypochlorite and a final rinse in water [38]. Additionally, faeces and urine were collected into sterile sample pots. In instances where the bladder was empty, a piece of the bladder wall was collected instead. Whole blood was collected from the heart into plain serum vacutainers. Blood samples were kept refrigerated until the end of the sampling, then they were centrifuged, and serum was separated prior to freezing [26].

## Nucleic acid extraction

DNA was extracted from approximately 100 mg of faeces using the Isolate II Fecal DNA Kit (Meridian Bioscience Inc., USA), following the protocol specified by the manufacturer. Tissues, serum and urine were extracted using the Real Genomics HiYield Genomic DNA Mini Kit for blood, bacteria and cultured cells (Real Biotech Corporation, Taiwan). In all cases, a negative extraction control consisting of 200 μL phosphate-buffered saline (PBS) in place of the sample was included with each extraction batch. Additionally, an exogenous internal control in the form of a known concentration of suspended *Listeria innocua* culture was added to all samples prior to extraction to determine the extraction efficacy or the presence of any inhibitors in the PCR. The manufacturer's protocol for frozen whole blood was used to extract DNA from 200 μL of serum and urine. The same protocol was used for tissues but with some modifications. Briefly, 40 mg of tissue was minced in a sterile petri dish using a disposable sterile scalpel blade in a biosafety cabinet, before transfer to a microcentrifuge tube. Samples were then incubated at 60°C on a shaker overnight in a suspension of 200 μL PBS, 200 μL GT buffer and 15 μL proteinase K, before continuing with the original protocol. This protocol modification was evaluated before the commencement of the study, as follows: tissue from a previously identified *C. burnetii* negative kangaroo was spiked with a known quantity of *C. burnetii* Nine Mile RSA439 (Phase II, Clone 4), which was obtained after repeated passage in Vero cells. DNA was extracted using varying quantities of tissue and reagents to determine the extraction efficacy. The extracted samples were tested for *C. burnetii* DNA by PCR targeting the *com1* gene and the extraction efficacy for each sample was determined by comparing the cycle threshold (Ct) values against a blank (PBS) control spiked with the same amount of culture.

## Real-time PCR detection

Real-time PCR reactions were performed at the Australian Rickettsial Reference Laboratory, using previously published protocols to individually detect the multicopy insertion sequence IS*1111*, the heat shock operon *htpAB* and outer membrane protein *com1* genes of the *C. burnetii* genome (S1 Table). PCR reactions were performed as previously described in 25 μL reactions, using a magnetic induction cycler (MIC) [39–41]. All samples were initially screened using the PCR targeting the IS*1111* gene, which is considered the most analytically sensitive due to its high copy number [42]. Specimens with no amplification after 40 cycles were considered negative, while those with a Ct value of < 40 cycles underwent confirmatory testing with a repeat PCR for IS*1111*, as well as primers targeting *htpAB* and *com1*. For a specimen to be classified as positive, it had to yield a Ct value of < 40 on both the initial screening test and at least one of the three follow-up PCRs. Kangaroos were classed as positive for *C. burnetii* if at least one tissue was confirmed positive by PCR.

Extraction success and possible PCR inhibition were measured in all samples and extraction controls by performing a PCR for *L. innocua*, targeting a 62 bp DNA fragment from the *lin02483* gene (S1 Table) [43]. An increase in Ct value of between 3–6 cycles compared to the PBS extraction control was considered indicative of moderate inhibition in the sample, while

an increase of > 6 Ct values was interpreted as possible severe inhibition [44]. A failure to amplify *Listeria* in a sample was considered indicative of extraction failure, and the sample was re-extracted. A no template control and relevant positive controls (*C. burnetii* Nine Mile RSA439 or *L. innocua* culture) were included in all PCR assays.

## Serology

All serum samples were tested for antibodies to *C. burnetii* phase I and II antigens using a macropod-specific indirect immunofluorescence assay (IFA) as part of the diagnostic test validation process, which has been reported elsewhere [26]. Briefly, serum was tested in duplicate at a 1:32 dilution in 2% casein PBS, using 40-well microscope slides coated with phase I or II *C. burnetii* antigen (Virion\Serion, Würzburg, Germany). Following incubation at 37°C for 40 minutes, a custom fluorescein-labelled rabbit anti-kangaroo IgG antibody diluted 1:200 in 2% casein PBS was added to each well and the incubation step was repeated. Slides were then washed, air-dried and mounted before they were examined for fluorescence with an immuno-fluorescence microscope at ×40 magnification. Positive and negative controls were included on each slide. Positive samples were subsequently titrated in serial 2-fold dilutions to determine their endpoint titres, which was defined as the maximum dilution in which both replicates exhibited strong fluorescence.

## Statistical analysis

Statistical analysis and data visualisation was performed in R, version 4.3.1 [45]. Fisher's exact test was used to test for associations between: (1) tissue type and PCR positivity; and (2) animal-level PCR positivity and serological status. In the remainder of this paper, we use the term prevalence to describe the proportion of tissues and/or animals positive to a given test result as we believe that the population of kangaroos sampled was representative of the population of kangaroos at risk. Molecular and serological *C. burnetii* apparent animal-level prevalence was calculated for categories within each demographic variable (sex, age, and species). The prevalence estimates for a given level of interest were then compared with an assigned reference category to return a prevalence risk ratio (RR). RRs were then adjusted for known confounders using the Mantel-Haenszel technique [46]. The statistical significance of the Mantel-Haenszel adjusted RR was tested using the chi-squared test and interpreted at the 5% level of significance.

## Results

A total of 537 specimens were collected from 50 animals: 27 eastern grey kangaroos and 23 red kangaroos, the majority female (*n* = 37). Most of the animals were adults (*n* = 39), while 11 were classed as sub-adults (Table 1). All the kangaroos appeared in good body condition, except for one aged female. Although efforts were made to obtain a complete set of tissue samples from each animal, mediastinal lymph nodes (*n* = 6), reproductive tissue (*n* = 4) and bladder wall/urine (*n* = 2) were not available in a minority of cases. There was also an insufficient volume of serum available from one animal, which precluded PCR analysis of one sample.

Of the 537 specimens tested, 179 were positive on the initial screening PCR with IS*1111*. Of these, 99 specimens from 42 kangaroos were positive on confirmatory testing, all of which for the IS*1111* gene. Only 16 and four of these samples, from 16 individuals, amplified using the *com1* or *htpAB* primers, respectively. Heart tissue was the most frequently positive (34%) sample type tested, while epididymis and testes had the lowest rates of detection (Table 2). Overall, 42 kangaroos (84%, 95% confidence interval [CI], 71% to 93%) were classified as PCR positive for *C. burnetii* based on confirmed detection with IS*1111* in at least one tissue (range: 1 to 7).

**Table 1. Breakdown of the apparent prevalence and the 95% confidence intervals (CI) for the PCR and IFA between species, sex and age groups.**

| Strata | Total tested | PCR | | IFA | |
|---|---|---|---|---|---|
| | | *n* pos | % pos (95% CI) | *n* pos | % pos (95% CI) |
| **Species:** | | | | | |
| Red kangaroo (*Osphranter rufus*) | 23 | 20 | 87 (66, 97) | 13 | 56 (34, 77) |
| Eastern grey kangaroo (*Macropus giganteus*) | 27 | 22 | 82 (62, 94) | 16 | 52 (39, 78) |
| **Sex:** | | | | | |
| Male | 13 | 9 | 69 (39, 91) | 3 | 23 (5.0, 54) |
| Female | 37 | 33 | 89 (75, 97) | 23 | 62 (45, 77) |
| **Age group:** | | | | | |
| Adult | 39 | 33 | 85 (70, 94) | 25 | 64 (47, 79) [a] |
| Subadult | 11 | 9 | 82 (48, 98) | 1 | 9.1 (0.2, 41) |
| **Total** | **50** | **42** | **84 (71, 93)** | **26** | **52 (37, 66)** |

[a] The prevalence of *C. burnetii* seropositivity in adult macropods, after adjusting for the effect of sex, was 4.5 (95% CI 0.9 to 22) times greater in adults compared with sub-adults ($\chi^2$ test statistic 6.48; $df$ = 1; p = 0.011).

When basing the classification on detection with *htpAB* and/or *com1*, the individual kangaroo-level prevalence was 32% (95% CI 20% to 47%). There was no statistically significant unconditional association between the probability of an animal being PCR positive and species, age group or sex (Table 1), nor between PCR positivity and sample type (p = 0.148).

The Ct values in all amplifications were generally high, with a median of 36.12 (range 29.28–38.24) for IS*1111*, 38.02 (range 37.40–38.29) for *htpAB*, and 37.19 (range 35.18–38.18) for *com1*. Evidence of some potential inhibition was present in 229 of 537 (43%) samples, with possible marked inhibition present in 32 (6%). Signs of inhibition were seen most frequently in lymph nodes and spleen, which made up 53% of samples with a delayed *Listeria* Ct of > 6. However, *C. burnetii* was still amplified in six of these samples, including two samples using *com1* and *htpAb* primers.

**Table 2. Positivity rate and 95% confidence intervals (CI) for the IS*1111*, *com1* and *htpAB* PCRs by sample type tested.**

| Tissue—sample type | Total tested | IS*1111* | | *com1* or *htpAB* | |
|---|---|---|---|---|---|
| | | *n* pos | % pos (95% CI) | *n* pos | % pos (95% CI) |
| Heart | 50 | 17 | 34 (21, 49) | 2 | 4.0 (0.5, 14) |
| Lung | 50 | 9 | 18 (8.6, 31) | 2 | 4.0 (0.5, 14) |
| Mediastinal lymph node | 44 | 8 | 18 (8.2, 33) | 3 | 6.8 (1.4, 19) |
| Mesenteric lymph node | 50 | 10 | 20 (10, 34) | 1 | 2.0 (0.1, 11) |
| Spleen | 50 | 10 | 20 (10, 34) | 1 | 2.0 (0.1, 11) |
| Liver | 50 | 6 | 12 (4.5, 24) | 3 | 6.0 (1.3, 16) |
| Testes | 13 | 0 | 0.0 (0.0, 25) | 0 | 0.0 (0.0, 25) |
| Epididymis | 12 | 1 | 8.3 (0.2, 38) | 0 | 0.0 (0.0, 26) |
| Uterus | 35 | 6 | 17 (6.6, 34) | 1 | 2.9 (0.1, 15) |
| Pouch | 36 | 9 | 25 (12, 42) | 0 | 0.0 (0.0, 9.7) |
| Urine/bladder[a] | 48 | 10 | 21 (10, 35) | 1 | 2.1 (0.1, 11) |
| Faeces | 50 | 8 | 16 (7.2, 29) | 4 | 8.0 (2.2, 19) |
| Serum | 49 | 5 | 10 (3.4, 22) | 2 | 4.1 (0.5, 14) |
| *Total kangaroos* | *50* | *42* | *84 (71, 93)* | *16* | *32 (20, 47)* |

Only samples that were confirmed positive in at least one follow-up PCR are included. All samples that amplified for *com1* and *htpAB* were also positive for IS*1111*.

[a] Three urine samples, 45 bladder wall samples.

**Table 3. Contingency table with the PCR and IFA results reported in this study.**

|  | IFA + | IFA - | *Total* |
|---|---|---|---|
| **PCR +** | 24 | 18 | 42 |
| **PCR -** | 2 | 6 | 8 |
| *Total* | 26 | 24 | 50 |

Kangaroos were defined as PCR positive if at least one sample type tested positive on the initial screening test (IS*1111*) and at least one of the follow-up tests.

Of the 50 kangaroos sampled, 26 (52%, 95% CI 37% to 66%) were positive for antibodies to *C. burnetii* on the IFA. Eight animals had antibodies to phase I only, while 18 had antibodies against both phase I and II [26]. Of the 26 seropositive kangaroos, 24 were also positive by PCR on at least one tissue, while only six animals were negative on all tests (Table 3). Titres ranged from 1:32 to 1:32,768 against both phases, with phase I antibodies generally being equal to or higher than phase II, except for two animals. Kangaroos with higher phase I titres ($\geq$ 1:8,192) were systematically PCR positive on at least one tissue and there was a general observation that seropositive animals tended to have the highest number of PCR positive tissues, although this was not statistically significant (p = 0.135; Fig 1). While the seroprevalence for females was greater (62%) than that of males (23%), most of the females that were sampled were adults, compared with an approximately equal number of adult and subadult males. After controlling for the confounding effect of age using the Mantel-Haenszel procedure, there was no statistically significant association between sex and seropositivity (RR = 1.58, 95% CI: 0.7 to 3.6; $\chi^2$ test statistic 1.537, p = 0.108; Table 1). The seroprevalence in adults (after adjusting for the effect of sex) was 4.5 times (95% CI: 0.9 to 22; $\chi^2$ test statistic 6.48; p = 0.011) that of subadults (Table 1).

Overall, the probability of detecting *C. burnetii* on a given test if a kangaroo was classed as PCR positive varied between samples and there was no single sample type that would reliably identify a positive animal. Of the 42 PCR positive kangaroos, serum antibodies were present in 24 (57%, 95% CI 41% to 72%), while heart tissue was positive in 17 animals (40%, 95% CI 26% to 57%; Fig 2). When interpreted in parallel, serology or PCR on heart tissue identified 30 (71%) of the 42 positive animals. PCR on the other tissues identified between 11 and 28% of the positive animals (Fig 2). The diagnostic sensitivity of the PCR to detect positive animals increased somewhat if the classification of positives was based on the initial IS*1111* screening test alone, although with the possible consequence of including false positives (Fig 3 and S2 Table).

## Discussion

The results from this study identified a surprisingly high apparent prevalence of *C. burnetii* amongst macropods in the study area. Moreover, *C. burnetii* appears to be widely distributed throughout a macropod's organs, as evidence of the pathogen was found in almost all the examined tissue types. Although a slightly higher overall rate of detection was found in heart tissue, there was no statistically significant difference in PCR positivity rate associated with sample type. Additionally, each tissue individually had a much lower positivity rate compared to the overall animal-level prevalence, meaning no one single tissue appears to be more suitable than others for identifying a *C. burnetii* positive macropod. Based on the results presented in this study, between 50% and 60% of PCR positive animals would likely be missed if only the most frequently positive organ, the heart, was sampled and tested. Similar findings, along with high Ct levels, have also been reported in roe deer (*Capreolus capreolus*), leading the author to

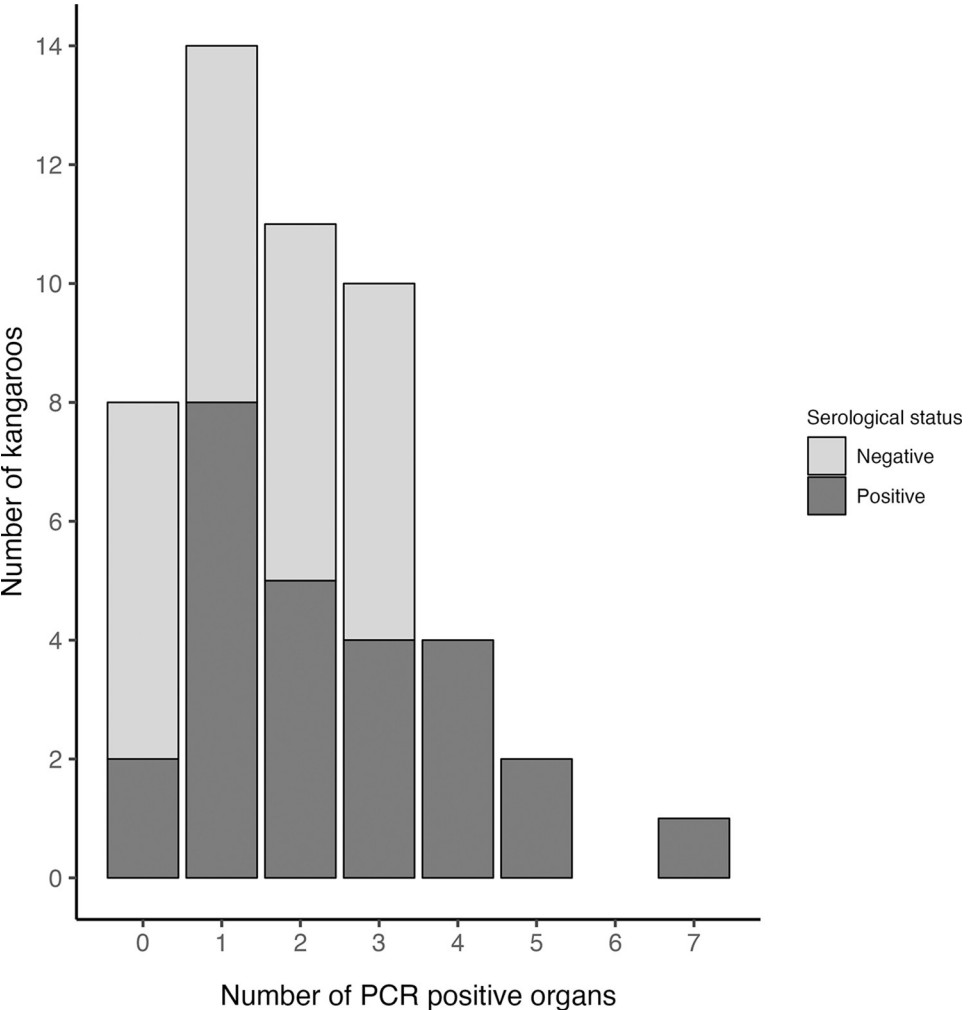

**Fig 1. Number of PCR positive tissues per kangaroo, by serological status.** Stacked bar plot showing the count of kangaroos as a function of the number of PCR positive tissues per kangaroo. Shading indicates serological status.

conclude that multiple tissues should be tested per animal to enhance the chance of detection [47]. This recommendation also appears to hold true for kangaroos, despite a relatively high overall apparent prevalence in these animals.

The high Ct values seen across all samples in this study presented a challenge for interpretation. This late amplification would indicate that the amount of *C. burnetii* DNA present in the tissues was generally low and possibly verging on the limit of detection, which would account for the lack of repeatability across assays in several of the samples. The multicopy insertion element (IS*1111*) is also considerably more sensitive than the single copy genes *com1* and *htpAB*, which helps explain the discrepancies in the detection rate between the PCR targets. This is evidenced by the difference in apparent prevalence between the assays, with an animal-level prevalence of 84% when using the duplicate IS*1111* compared with a more conservative estimate of 32% for *com1/htpAB*. The IS*1111* was therefore selected as the screening test in this study, although the high analytical sensitivity of this assay could predispose to the amplification of environmental or cross-contaminants, potentially resulting in false positives. While inadvertent DNA contamination of field samples cannot be fully excluded, all reasonable steps

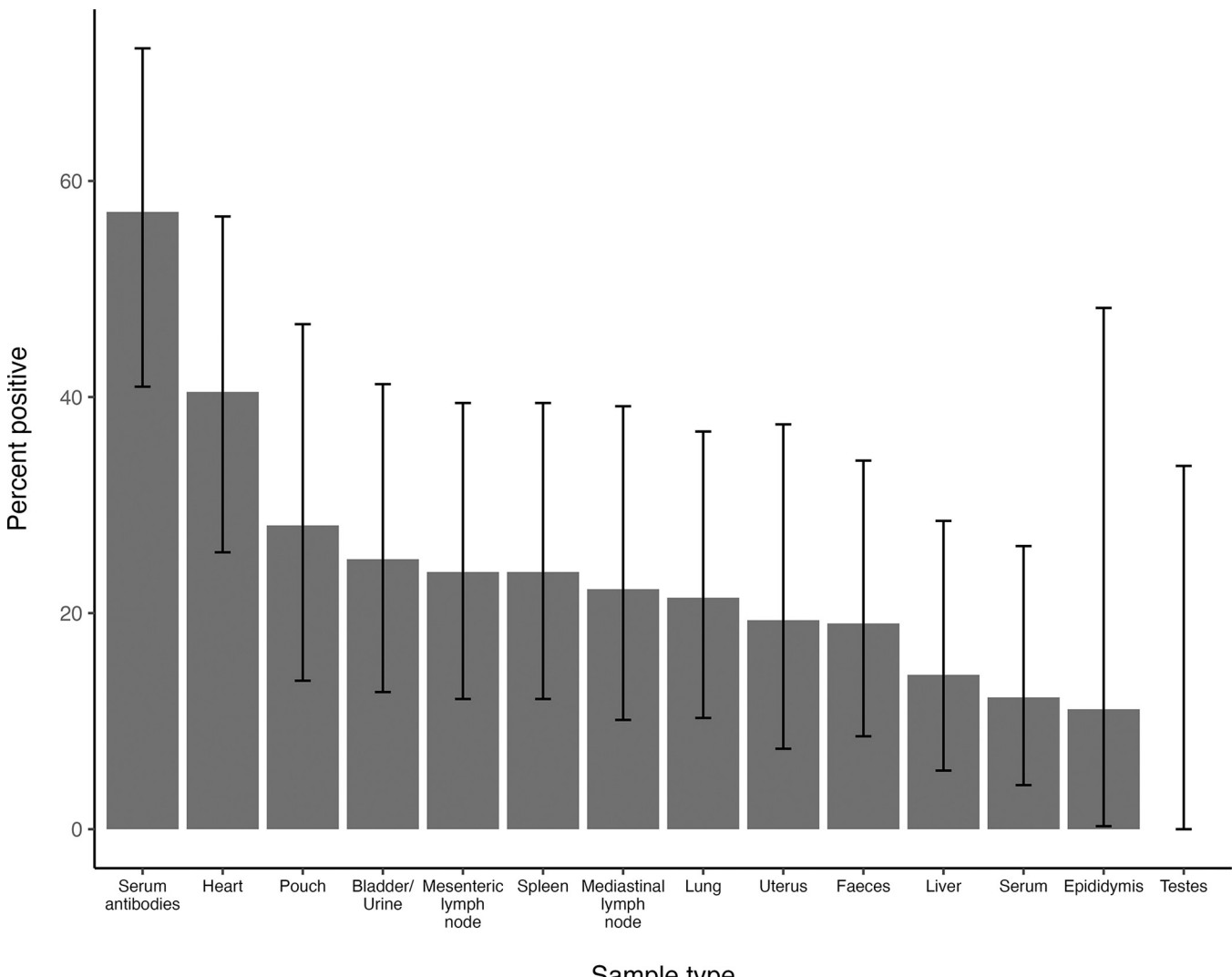

**Fig 2. Individual specimen positivity rate in confirmed PCR positive kangaroos.** Ranked bar plot showing the percentage likelihood of a given sample type being confirmed positive for *C. burnetii* in kangaroos that were PCR positive on at least one tissue. The seroprevalence in the same population, as determined by the immunofluorescence assay, is included for comparison. The error bars represent the 95% confidence intervals.

were taken to avoid cross-contamination during sample collection, storage, and processing. Furthermore, cross-contamination would be considered unlikely in the absence of any strongly positive samples to act as a source for contamination. However, to help mitigate this concern, we required *Coxiella* DNA to amplify in at least two assays for a sample to be classified as positive. Although most of the positive samples only amplified IS*1111*, multiple samples also amplified *com1* and *htpAB*, which lends credibility to the results. The results from this study indicate that employing the IS*1111* PCR as a screening test increases the likelihood of detecting *C. burnetii* in kangaroo tissues, although positive results should ideally be confirmed with follow-up testing.

Unfortunately, the high Ct values also meant that the samples were unsuitable for genotyping or isolation, which is a limitation of this study. Multilocus variable-number tandem repeat analysis (MLVA) has been used to reliably characterise Australian strains of *C. burnetii*, however, this requires samples to yield a lower Ct value to be successful [32,48,49]. Genotyping or

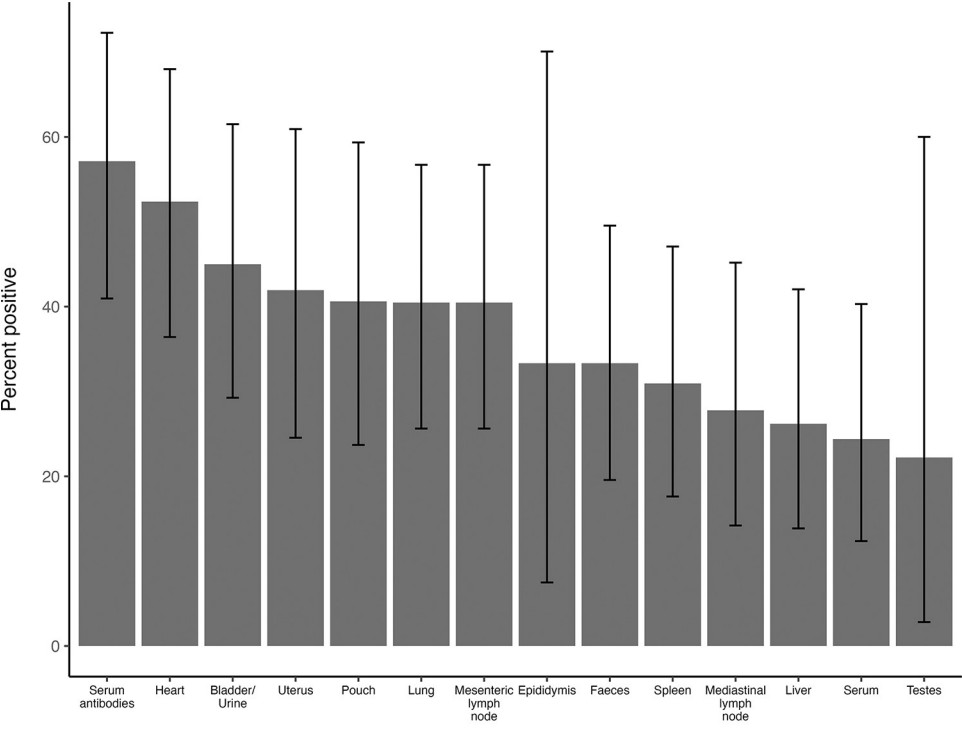

**Fig 3. Individual specimen detection rate based on the screening test alone in confirmed PCR positive kangaroos.**
Ranked bar plot showing the percentage likelihood of a given sample type being positive for *C. burnetii* on the initial screening PCR with IS*1111*, in the 40 kangaroos that were confirmed PCR positive on at least one tissue. The seroprevalence in the same population, as determined by the immunofluorescence assay, is included for comparison. The error bars represent the 95% confidence intervals.

sequencing of a suitable isolate from one of these kangaroos would allow comparison with other variants isolated from humans or livestock and would more accurately inform the extent to which these kangaroos contribute to the animal reservoir of Q fever. This confirmatory testing would also be necessary to confidently differentiate between *C. burnetii* and *Coxiella*-like endosymbionts, many of which possess variations of the IS*1111*, *com1* and *htpAB* genes [50–53]. It is conceivable that the potentially high level of inhibition observed in some samples, as determined by the exogenous control, contributed to the late amplifications of *C. burnetii*. However, it is important to note that the high Ct values were consistent across all positive samples, including those with no or minimal delays in the *Listeria* Ct values. Furthermore, even samples showing signs of potential inhibition still successfully amplified for *C. burnetii*, so the actual impact of the suspected inhibition remains uncertain. Nonetheless, it is possible that the identification and elimination of any remaining inhibitors could result in reduced Ct values or an increase in the number of amplifications.

The transmission routes of *C. burnetii* from macropods are poorly understood. Although there was evidence of *C. burnetii* being present in uterine tissue in this study, large-scale shedding through birth products, such as seen in placental mammals, is unlikely to be a feature in marsupials due to their different physiology and lack of placenta. Sexual transmission between kangaroos, as has been demonstrated in rodents, cannot be ruled out but is perhaps of lesser importance based on the low level of detection in male reproductive tissues in this study [54]. Conversely, the identification of *Coxiella* DNA in 16% of faecal samples and 21% of urinary

tract samples suggests that kangaroos possibly excrete the bacteria in their faeces and urine, although it remains uncertain whether this finding signifies the presence of viable bacteria or if it simply represents contamination with genetic material. Confirmation of this would be important and could be achieved through successful culture and isolation of the bacteria from either urine or faeces. The detection rate in faecal samples in this study is consistent with previous studies in macropods and other animals [23,24,55,56]. Similarly, *C. burnetii* DNA has also been reported in the urine of flying foxes, koalas, domestic carnivores, horses, ruminants, and dromedary camels [23,55,57]. Although the amount of genetic material detected in these samples appeared to be low, it may still contribute to transmission, given the infectious dose of *C. burnetii* can be as low as a single organism [58].

Unfortunately, no samples from the kidneys or the gastrointestinal tract were available for testing in this study. Considering the relatively frequent detection in the bladder/urine and faecal samples, it would be useful to include these tissues in future surveys, particularly as *C. burnetii* has been shown to persist for a long time in kidneys in some experimentally infected mammals and birds [59–61]. Histopathology and immunohistochemistry on fixed tissues from positive animals could also be useful to further characterise *C. burnetii* infections in macropods and may shed more light on the pathogen-host relationship, such as whether there is any particular tissue tropism or if the infection is associated with pathology.

The study population showed little congruence between PCR positivity and serological status, however, a large proportion of kangaroos (48%) tested positive by both PCR and the IFA. In a previous study that compared serology and faecal PCR in western grey kangaroos, the authors found that the likelihood of detecting *C. burnetii* DNA in faeces was seven times greater if the animal was also seropositive [24]. As determining the stage of an infection through a cross-sectional survey is not possible, the concurrent presence of both the pathogen and antibodies in the same animal could indicate either a recently established, persistent, or repeated infection. While the immune response to *C. burnetii* in kangaroos is largely unknown, it has been studied extensively in humans, laboratory animals and domestic ruminants, where it has been found to be complex [62–64]. In these species, the cellular immune response is considered the most important component for infection clearance, while antibodies appear to be dispensable [14,63]. The presence of antibodies alone is therefore unlikely to be sufficient to clear an infection. Although it is difficult to say how this compares to the marsupial immune system, the large overlap between PCR positivity and seropositivity might be attributed to a similar dynamic, particularly if there is also substantial infection pressure.

The relative distribution of phase I and II antibodies seen in this study population is also noteworthy. In experimental infection studies on other species and in naturally infected people, phase II antibodies are typically produced first, reach higher peak levels and may persist for longer than antibodies to phase I [64–68]. In most animal surveys, phase II antibodies therefore tend to occur more frequently than phase I [69–72], however, the opposite was observed in the kangaroos included in this study. Similar findings have been consistently reported in other macropod populations [18,26], with the exception of one large-scale study using an unvalidated competitive enzyme-linked immunosorbent assay, where the relative antibody distribution varied with sampling location [25]. This reverse pattern appears to be a peculiar feature common in macropods, although it has also occasionally been documented in Steller sea lions (*Eumetopias jubatus*), harbour seals (*Phoca vitulina*), snowshoe hares (*Lepus americanus*), white rhinoceros (*Ceratotherium simum*) and moose (*Alces alces*) [73–76]. High antibody titres to phase I correspond with persistent Q fever in humans but this humoral response is ineffective in clearing the infection [14]. A similar phenomenon could potentially occur in kangaroos, as animals with high phase I titres were also PCR positive, and a higher number of PCR-positive organs were found in seropositive animals. However, there is

currently no evidence that *C. burnetii* causes similar disease in macropods. Understanding how and why the antibody kinetics differ in these species would likely necessitate longitudinal surveys with sequential sampling or experimental infection studies, which is challenging in wildlife.

Based on the results from this study, recommendations for sampling macropods for Q fever will depend on the desired objective. For surveillance purposes, where the aim is to detect evidence of *C. burnetii* in a population, the most time- and resource-efficient option would likely be serosurveys using the IFA. This could be combined with molecular testing targeting the IS*1111* gene to increase sensitivity or to differentiate between current infection and past exposure. However, consideration must be given to sample sizes and the range of tissues available to account for the seemingly low concentration and patchy distribution of the organism within tissues. PCR on faeces allows for a non-invasive alternative screening tool but is insensitive on its own and would require a larger sample size to ensure detection. Any positive PCR results should be confirmed with follow-up testing. Conversely, if the aim is to detect coxiellosis in an individual animal post mortem, the chances of detection would be maximised by sampling and testing as many tissues as possible. If a diagnosis is sought in a live kangaroo, or if an animal needs to be declared free of infection (for example pre- or post-translocation of a captive animal), paired serology using the IFA and sequential molecular testing of faeces and urine would be recommended.

## Conclusion

There is increasing evidence that macropods play some part in the epidemiology of Q fever in Australia. Based on the results from this study, *C. burnetii* appears to be present in a high percentage of eastern grey and red kangaroos in the study area, albeit seemingly with an inconsistent distribution within tissues and in relatively small quantities, often verging on the limits of detection. Due to this, testing protocols for coxiellosis in macropods should be tailored to the desired objective. Shedding of the organism in urine and faeces appears to be a likely feature of infection in macropods and could be a route of transmission to other animals or people, although the relative contribution of macropods to the reservoir of *C. burnetii* remains stubbornly elusive.

## Supporting information

**S1 Table. Primers used in real-time PCR reactions in this study.**
(DOCX)

**S2 Table. The number of kangaroos and distribution of tissues that were positive for C. burnetii on the initial screening PCR for IS1111.**
(DOCX)

## Acknowledgments

We acknowledge the contributions of Dr Nicholas J. Clark (the University of Queensland), Chelsea Nguyen (Australian Rickettsial Reference Laboratory) and Dr. Michael Muleme (Victorian Government Department of Health).

## Author Contributions

**Conceptualization:** John Stenos, Charles Caraguel, Mark A. Stevenson.

**Data curation:** Anita Tolpinrud.

**Formal analysis:** Anita Tolpinrud, Mark A. Stevenson.

**Funding acquisition:** John Stenos, Charles Caraguel, Mark A. Stevenson.

**Investigation:** Anita Tolpinrud, Mythili Tadepalli.

**Methodology:** Anita Tolpinrud, John Stenos, Joanne M. Devlin.

**Resources:** John Stenos, Louis Lignereux.

**Supervision:** John Stenos, Anne-Lise Chaber, Joanne M. Devlin, Mark A. Stevenson.

**Validation:** John Stenos.

**Visualization:** Anita Tolpinrud.

**Writing – original draft:** Anita Tolpinrud.

**Writing – review & editing:** Mythili Tadepalli, John Stenos, Louis Lignereux, Anne-Lise Chaber, Joanne M. Devlin, Charles Caraguel, Mark A. Stevenson.

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
