## [Decision Letter · Decision Letter 0]

19 Feb 2024

PONE-D-23-43149Tissue distribution of *Coxiella burnetii* and antibody responses in macropods co-grazing with livestock in Queensland, AustraliaPLOS ONE

Dear Dr. Tolpinrud,

Thank you for submitting your manuscript to PLOS ONE. After careful consideration, we feel that it has merit but does not fully meet PLOS ONE’s publication criteria as it currently stands. Therefore, we invite you to submit a revised version of the manuscript that addresses the points raised during the review process.

We look forward to receiving your revised manuscript.

Kind regards,

Gianmarco Ferrara, PhD, MVD

Academic Editor

PLOS ONE

Journal Requirements:

Reviewers' comments:

Reviewer's Responses to Questions

**Comments to the Author**

1. Is the manuscript technically sound, and do the data support the conclusions?

Reviewer #1: Partly

Reviewer #2: Yes

2. Has the statistical analysis been performed appropriately and rigorously? 

Reviewer #1: Yes

Reviewer #2: Yes

3. Have the authors made all data underlying the findings in their manuscript fully available?

Reviewer #1: Yes

Reviewer #2: Yes

4. Is the manuscript presented in an intelligible fashion and written in standard English?

Reviewer #1: Yes

Reviewer #2: Yes

5. Review Comments to the Author

Reviewer #1: 1. In this paper, the authors report results of a cross sectional survey of C. burnetii in macropods in Australia. Opportunistic sampling of 50 animals was done and different tissue samples were collected over a 3 day period.

2. The study seem to focus on i) diagnostics; comparing PCR detection frequencies from different sample types as well as suitability of three PCR markers for C. burnetii detection. The PCR results were also evaluated in comparison with serology results that were already reported somewhere else (ref 23). My criticism is that some of the principles of assay development/ optimisation were omitted. ii) The apparent sero detection as well as the molecular detection at animal level (referred to as apparent prevalence, with n=50 animals tested), including demographic data association. My criticism is that, the serology data seem to have been reported elsewhere and it is not clear why were these results included in this part of the study.

Reviewer #2: The manuscript "Tissue distribution of Coxiella burnetii and antibody responses in macropods co-grazing with livestock in Queensland, Australia" is well-written and describes the occurrence of Cb DNA in different organs/samples from kangaroos in Queensland. Moreover, blood samples were screened for antibodies.

I have only minor comments:

General comment: You talk about "prevalence" throughout the entire manuscript. In my opinion, you did not conduct a classical prevalence study. Your sample size follows a convenient sample approach. Therefore, I suggest using terms such as "detection rate" or "positivity rate" instead of "prevalence."

Line 66: "... promote transmission." Please insert a reference from the literature.

Line 120: Depending on the journal, but I suggest including the ethics statement at the end of the manuscript.

Lines 421-422: "... to occur more frequently than phase I." You cited very old literature. In recent years, newer findings have been published about the detection of Cb phase-specific antibodies in animals. Please revise the citations.

Table 1: Please include the English names of both kangaroo species.

6. PLOS authors have the option to publish the peer review history of their article (what does this mean?). If published, this will include your full peer review and any attached files.

Reviewer #1: **Yes: **Dr Nomakorinte Gcebe

Reviewer #2: No

---

## [Author Response · Author response to Decision Letter 0]

19 Mar 2024

Reviewer #1: 

Comment: “The study seem to focus on i) diagnostics; comparing PCR detection frequencies from different sample types as well as suitability of three PCR markers for C. burnetii detection. The PCR results were also evaluated in comparison with serology results that were already reported somewhere else (ref 23). My criticism is that some of the principles of assay development/ optimisation were omitted.”

Author response: Thank you for this comment. The IFA assay development and validation was previously published in detail in the below citation and is therefore only referenced in the current manuscript rather than repeated: 

Tolpinrud A, Stenos J, Chaber AL, Devlin JM, Herbert C, Pas A, et al. Validation of an Indirect Immunofluorescence Assay and Commercial Q Fever Enzyme-Linked Immunosorbent Assay for Use in Macropods. J Clin Microbiol. 2022:e0023622. Epub 20220602. doi: 10.1128/jcm.00236-22

We have modified the text slightly (line 206-207) to make this clearer. Similarly, the PCR assays used here have been published previously (references 39-41) and are routinely used in the Australian Rickettsial Reference Laboratory for diagnostic and research purposes. They were therefore not developed or optimised as part of this study. 

Comment: “The apparent sero detection as well as the molecular detection at animal level (referred to as apparent prevalence, with n=50 animals tested), including demographic data association. My criticism is that, the serology data seem to have been reported elsewhere and it is not clear why were these results included in this part of the study.”

Author response: Thank you for this comment. As mentioned in lines 112-117, only the overall seroprevalence rate (i.e. the crude apparent and estimated true seroprevalence) is reported elsewhere, while the current manuscript details the antibody distribution in relation to phases and demographics (age/sex/species). The referenced article (ref. no. 26) details the assay development and validation process, a by-product of which is the estimated seroprevalences for the tested kangaroo populations. A detailed breakdown of relative titres and phase variations, as is given in the current manuscript, is not included in the previous paper. Additionally, the current manuscript presents a unique opportunity to report on the serological results in the context of the molecular detection of C. burnetii, which was also not available in the cited article. 

The reason this is highlighted in the text is to ensure transparency so readers are aware that the sample set detailed in this text is the same as the Roma population in reference 26. 

Reviewer #2: 

Comment: “General comment: You talk about "prevalence" throughout the entire manuscript. In my opinion, you did not conduct a classical prevalence study. Your sample size follows a convenient sample approach. Therefore, I suggest using terms such as "detection rate" or "positivity rate" instead of "prevalence."”

Author response: Thank you for this comment. It is our assessment that the kangaroos that were sampled were representative of the population of kangaroos at risk. For this reason, ‘prevalence’ is the appropriate term, acknowledging of course, that given a relatively small sample size the precision of our prevalence estimate(s) is not high. See lines 221 to 224 in the revised manuscript.

Comment: “Line 66: "... promote transmission." Please insert a reference from the literature.”

Author response: The manuscript has been updated with three relevant references (references 11-13).

Comment: “Line 120: Depending on the journal, but I suggest including the ethics statement at the end of the manuscript.”

Author response: Thank you for this comment. We followed the PLOS One author guidelines for the reporting of animal ethics, which state: “Manuscripts reporting animal research must state in the Methods section: The full name of the relevant ethics committee that approved the work, and the associated permit number(s). Where ethical approval is not required, the manuscript should include a clear statement of this and the reason why. Provide any relevant regulations under which the study is exempt from the requirement for approval.” 

Comment: “Lines 421-422: "... to occur more frequently than phase I." You cited very old literature. In recent years, newer findings have been published about the detection of Cb phase-specific antibodies in animals. Please revise the citations.”

Author response: The references have been updated by removing the oldest citation and adding two more recent publications (references 63-68). We would welcome suggestions if the reviewer has any particular citations in mind. 

Comment: “Table 1: Please include the English names of both kangaroo species.”

Author response: Thank you for this comment. The table has been updated as requested.

---

## [Decision Letter · Decision Letter 1]

2 May 2024

Tissue distribution of *Coxiella burnetii* and antibody responses in macropods co-grazing with livestock in Queensland, Australia

PONE-D-23-43149R1

Dear Dr. Tolpinrud,

We’re pleased to inform you that your manuscript has been judged scientifically suitable for publication and will be formally accepted for publication once it meets all outstanding technical requirements.

Kind regards,

Gianmarco Ferrara, PhD, MVD

Academic Editor

PLOS ONE

Additional Editor Comments (optional):

Reviewers' comments:

Reviewer's Responses to Questions

**Comments to the Author**

1. If the authors have adequately addressed your comments raised in a previous round of review and you feel that this manuscript is now acceptable for publication, you may indicate that here to bypass the “Comments to the Author” section, enter your conflict of interest statement in the “Confidential to Editor” section, and submit your "Accept" recommendation.

Reviewer #2: All comments have been addressed

2. Is the manuscript technically sound, and do the data support the conclusions?

Reviewer #2: Yes

3. Has the statistical analysis been performed appropriately and rigorously? 

Reviewer #2: Yes

4. Have the authors made all data underlying the findings in their manuscript fully available?

Reviewer #2: Yes

5. Is the manuscript presented in an intelligible fashion and written in standard English?

Reviewer #2: Yes

6. Review Comments to the Author

Reviewer #2: All comments have been addressed. The manuscript is ready for publication. No further modifications are necessary.

7. PLOS authors have the option to publish the peer review history of their article (what does this mean?). If published, this will include your full peer review and any attached files.

Reviewer #2: No

---

## [Editor Report · Acceptance letter]

9 May 2024

PONE-D-23-43149R1 

PLOS ONE

Dear Dr. Tolpinrud, 

I'm pleased to inform you that your manuscript has been deemed suitable for publication in PLOS ONE. Congratulations! Your manuscript is now being handed over to our production team.

Kind regards, 

on behalf of

Dr. Gianmarco Ferrara 

Academic Editor

PLOS ONE